# Oral Health Status in a Group of Roma Children in Seville, Spain

**DOI:** 10.3390/healthcare11071016

**Published:** 2023-04-03

**Authors:** Ana Raquel García-Barata, Irene Ventura, David Ribas-Pérez, Javier Flores-Fraile, Antonio Castaño-Séiquer

**Affiliations:** 1Department of Paediatric Dentistry, Egas Moniz School of Healht & Science, 2829-511 Almada, Portugal; 2Department of Stomatology, Universidad de Sevilla, 41004 Sevilla, Spain; 3Department of Surgery, Universidad de Salamanca, 37008 Salamanca, Spain

**Keywords:** social dentistry, epidemiology, ethnic minorities, social disadvantages

## Abstract

The Spanish gypsy community is widely integrated among the rest of the Spanish population due to a coexistence that dates back centuries. Despite this integration, they are at risk of marginalization, the child population clearly being a vulnerable group. In terms of social and health inequalities, ethnic minorities in general, and the gipsy minority in particular, in many cases do not achieve equity with the rest of the population. Regarding health in general and, more specifically, oral health, this fact can be perceived, although it has not been evidenced by any epidemiological study of oral health in the Andalusia region. Objective: Identify the oral health status through an epidemiological survey of the population of gipsy children in the city of Seville. Methods: The WHO (World Health Organization) criteria for oral health surveys were used in the study with children aged 6 to 13 years carried out in two Seville schools located in neighborhoods with a high percentage of gypsy population (Polígono Norte and Sur of the city of Seville). Results: The final sample consisted of 108 children in whom the DMF (decay-missing-filled index) for primary dentition was 5.0 + −3.1 for the 6–9 year-old age group and the DMFT (Decayed, Missing, and Filled Teeth) for the 10–13 year-old group was 4.5 + −3.3. The frequency of brushing was low, with a significant percentage of children not brushing their teeth (42.3%). The striking feature is that 26.9% of children had never visited the dentist despite their basic dental care being covered by the regional government. Conclusions: The children studied present high levels of caries compared to the rest of the Spanish population, as well as very low levels of oral hygiene. Given the lack of success of existing oral health programmes among this population, a different intervention is needed, taking into account the idiosyncrasies of the gipsy community.

## 1. Introduction

The social determinants of health (SDH), in accordance with WHO, are the nonmedical elements that affect health outcomes. In addition to the larger group of factors and systems influencing the conditions of daily life, these are the circumstances in which people are born, develop, work, live, and age. These factors and systems include political systems, social norms, social policies, economic policies and systems, and development objectives.

Health inequalities, or the unfair and preventable variations in health status seen within and between regions, groups of population, or even countries, are significantly influenced by SDH. Health and sickness follow a social gradient across these regions of all income levels: the worse the socioeconomic standing, the worse the health [1].

On the other hand, health equity, as defined by the WHO, is the absence of unjust and preventable or redressable disparities in health among population groups classified according to social, economic, demographic, or geographic factors. Both SDH and equity in health are important aspects to study in disadvantaged populations, and this includes not only health in general but also, very importantly, oral health in particular.

This is where social dentistry finds its purpose. Scientific evidence shows a close parallelism between social inequalities and lack of oral health [1,2,3,4,5]. 

In recent years, there have been many social dentistry programmes for excluded populations, with a clear preventive vocation in their application and development, in line with the definition of health provided by WHO as a full biological–psychological condition and also determined by sociocultural determinants [1].

One might think that this excluded population is limited to underdeveloped or third-world countries, but the so-called first world also presents situations and groups of clear social exclusion. The famous University of Harvard in Massachusetts (USA) has developed social programmes in families with low income power, homeless people [6,7,8], Spanish-speaking minorities [9,10,11], and the disabled [12], where social dentistry plays an important role. In Spain, the social dentistry activities of the Valencia and Seville Universities with specific programmes for vulnerable populations stand out [2,4].

One of the minorities that has classically stood out for its lower socio-economic status and therefore presenting a socio-health situation that needs to be analysed is the Roma ethnic group. The Roma community in Spain is the largest ethnic minority of European origin, only surpassed in number by the Romanian community. This population is estimated to be between 800,000 and 1.5 million people. Roma people are considered the most excluded sector of the Spanish society. Due to this situation, it is evident that they constitute a group with special health needs, and are therefore susceptible to receive the benefits of social dentistry [2,13,14,15,16].

Severe poverty is a factor affecting the majority among the Spanish Roma population. According to data from the Fundación Secretariado Gitano, in 2007, it affected 69.6% of the Roma population, and in 2013, it affected 71.1% [15,16]. 

The Roma community also suffers from a situation of health inequality compared to the rest of the Spanish population. In a comparison of the 2006 National Health Survey of Spain with the same survey applied to the Roma population, the latter has a higher prevalence of chronic diseases, a higher prevalence of vision and hearing problems, a higher percentage of male smokers, lower daily consumption of fruit and vegetables, higher consumption of saturated fats, higher levels of overweight and obesity, etc., and less access to oral health services. It is particularly striking that half of the Roma child population has never visited a dental clinic [17].

The highest percentage of the Roma population resides in Andalusia, which comprises more than 40% of the Spanish Roma population. A significant percentage of the population are in a situation of social exclusion, residing in the so-called "Areas in Need of Social Transformation" [17,18,19]. Among the provinces, Seville has the highest percentage of Roma inhabitants (23.69%) with respect to the total Andalusian Roma population, and Granada has the highest proportion of Roma inhabitants with respect to the non-Roma population (7.53%) [17].

The aim of the study was to identify the oral health status of a group of school children aged 6 to 13 years belonging to the Roma population in Seville who are at risk of social exclusion. We want to provide this data to the responsible administrations in order to highlight their wide-ranging needs for improvement in their oral health.

## 2. Materials and Methods

### 2.1. Study Type and Settings

The design of the present study is cross-sectional. Data were obtained through a survey during the last week of September in 2019 that included an examination of the oral cavity of the study group with standardised criteria. There was also a questionnaire with specific questions about oral health. This questionnaire was validated by WHO as it is part of the book *Oral Health Surveys. Basic Methods*. More concretely, the annex 8 questionnaire was used [20].

School children were selected from two public schools, one located in the Polígono Norte and the other in the Polígono Sur of the city of Seville. The study was supported by Fundación Odontologia Social (FOS) with links to the University of Seville as well as public administrations of Andalusia.

These "Areas in Need of Social Transformation" concentrate all the characteristic problems that define a group in a situation of social exclusion and poverty, according to the conclusions of the Employment and Social Policy Council on the dimensions of social exclusion: dysfunctional families, low income, high unemployment, addictions, need for housing and public services. 

### 2.2. Study Variables and Criteria

The World Health Organisation (WHO) criteria were used for dental caries, periodontal disease, malocclusion, fluorosis, and care needs related to the condition of the teeth, as described in their Oral Health Surveys book [20].

The school children were examined in natural light and a No. 5 flat mirror was used; the teeth were not dried prior to inspection. All patients had the same clinical examiner. To measure the consistency of the observations, the intraobserver calibration was measured, obtaining a percentage of agreement with the Kappa test (0.80).

A sample size calculation was performed for the average caries value reported in the oral health survey conducted in 2015 in Spain [21]. For this calculation, a standard deviation of 1.8 and a precision level δ = 0.5, a type I error α = 0.05 and a power (1 − β) = 0.80 were used. The total estimated sample was 104 school children. 

### 2.3. Statistical Analyses

Comparison between sex within age groups with respect to quantitative variables used the T student test, while the Chi-square test was used to compare the influence of sex with respect to qualitative variables. A p value of *p* < 0.05 was established to determine a statistically significant difference. SPSS v.21.0 software (Statistical Package for Social Sciences, Chicago, IL, USA) was used for all analyses.

## 3. Results

### 3.1. Demographic Aspects 

The total number of participants was 108 school children, with a 50/50 gender distribution. The mean age was 9.6 (±2.0) years. 

The children were grouped into two age groups according to their division within the school in their classes and other activities: 6–9 years and 10–13 years distributing the children in a similar percentage between the ages of 10 and 13 years (51.9%) and between 6 and 9 (52.48%). The area of residence of the children examined was Seville south (81.5%) and the rest originated from Seville north (18.5%).

The occupation of the father of the family showed that most of them were street vendors in both age groups (88.5% in the 6–9 years and 73.2% in the 10–13 years). The percentage of unemployed parents was 9.25% in both age groups. The presence of parents engaged in untrained manual activities was also detected (Table 1).

### 3.2. Dental Caries

The prevalence of total caries in the sample for the primary dentition was 70.37% and for the permanent dentition it was 61.11%. 

If we analyse the DFT and DMFT indexes, the average number of caries in the primary dentition at the age of 6–9 years was higher (DFT 5.0 ± 3.1) than in the age group of 10–13 years (DFT 1.3 ± 2.0).

In the permanent dentition, the average number of decayed teeth is higher in the age group of 10–13 years (DMFT 4.5 ± 3.3) than in the 6–9 year-age group (DMFT 1.2 ± 1.4). 

More striking is the percentage of restorations (Restorations Index -RI-), which is non-existent in the primary dentition in the two age cohorts studied and which only reaches 0.6% in the permanent dentition in the entire sample for an extremely high prevalence of caries.

In the analysis of SiC (caries significance index), which refers to the average caries index of the third most prevalent disease in the studied population, the figure rises to 10.2 (Table 2).

Table 3 shows the dental treatment needs in the age subgroups of the sample. Children aged 6–9 years had on average more teeth in need of extraction and endodontics (7.0 ± 2.8) than the older group (3.3 ± 2.6). On the other hand, in children aged 10–13 years the most prevalent treatment need are fillings (6.8 ± 2.3), with a higher average than that observed in the group aged 6–9 years (3.3 ± 2.5).

Most of children aged 6–9 years needed some fillings (76.9%) and three or more invasive interventions (65.4%). In the older children, the majority needed three or more fillings (71.4%) and three or more invasive treatments (43.5%) such as endodontics or extractions.

### 3.3. Periodontal Status 

Most of the sextants had some need for treatment, mainly brushing instructions (CPI = 1) due to gingivitis. On average, only 2.4 ± 2.6 sextants were healthy at the time of examination according to the CPI. Furthermore, on average, 1.0 ± 1.9 sextants needed tartar removal prophylaxis. Tooth 1.6 had the highest calculus frequency (29.6%), while tooth 1.1 was the most prevalent bleeding tooth in the study group (48.1%). 

When assessing the needs of all the sextants, it was observed that all children explored the required periodontal intervention. Most (70.4%) would require prophylaxis and training in brushing techniques.

### 3.4. Malocclusions and Other Variables

Table 4 presents the malocclusion data for 12- and 13-year-olds. It was identified that 87.8% of the patients had some form of malocclusion, with the moderate/severe form being the most prevalent condition (63%). No child had fluorosis and the prevalence of molar incisor hypomineralisation (MIH) was 40.7%.

Regarding the presence of oral symptoms, 70.1% of subjects experienced pain in the last 12 months. A high percentage of children (48.1%) experienced pain and discomfort quite often or very often in the last 12 months. In the same period, 44.4% reported chewing problems occasionally or more frequently. In the analysed group, approximately two-fifths of the children had no problem eating and chewing in the last 12 months (59.3% of the school children).

### 3.5. Crossing of Variables

Table 5 presents the results on the frequency of brushing by sex. Girls were reported to brush their teeth more frequently than boys, this difference being statistically significant (*p* < 0.05). 

Two fifths of the boys do not brush their teeth at all (never 40.7%) and 48.1% brush less than once a week. On the contrary, the majority of girls brush once a week or less frequently (66.6%).

Table 6 shows the restorative treatment needs by age group and sex. It was observed that the need for fillings in the studied Sevillian gypsy girls aged 6–9 years was significantly higher (2.9 ± 2.4 teeth) than that of the boys (1.5 ± 1.4) (*p* < 0.05). This trend does not appear to be maintained in the age group of 10–13 years. Here, we obtained the value of *p* > 0.05.

Information on dental visits by sex is presented in Table 7. It was determined that approximately half of the boys have never visited the dentist and a quarter of the girls have never visited the dentist. This difference is not statistically significant (*p* > 0.05).

## 4. Discussion

One of the limitations of the study is that it is a cross-sectional survey, and it is like a photograph of the group studied. Analysing numerous correlations, we are unable to determine which of the "related" variables is the origin of or result of the other. Studies that use a cross-sectional design are more prone to bias. For instance, we can encounter a circumstance in which a variable that seems to be the cause of an event is really only an effect of the event.

Regarding the specificity of the strengths of the study, we believe that its main value lies in the fact that there is no similar study carried out in this population in Spain. 

Given that we have already analysed the importance of this ethnic minority and for the sake of their integration, this situation analysis can be a starting point to highlight the needs for specific oral health prevention programs for the studied population.

In the present study, both the prevalence of caries and dental caries rates among Roma children were high. Dental caries is the most prevalent oral disease among children and adolescents which increases progressively with age. It represents a major public health problem with great impact on the health and well-being of children and adolescents.

In a Spanish study conducted in the city of Granada among school children, caries was detected in 21.7% of cases [22], a much lower percentage than in our sample. If we compare the data obtained with epidemiological studies carried out in Spain, in the latest study carried out in 2020, we observe a similar prevalence (28.3% at 5 years of age and 11.6% at 12 years of age) to that of the aforementioned study and much lower than that of our study, bearing in mind that the same WHO criteria for the diagnosis of caries were used [23].

In other realities, for example, in Nepal, Prasai et al. analysed children in a primary school where the prevalence of caries in 5–6-year-olds was 52%, and in 12–13-year-olds it was 41% [24], a figure which, although high, is still lower than that obtained in the Roma children studied. Perhaps the socioeconomic context here is similar, but other parameters may be involved.

In Syria, Dashash et al. observed a 61% prevalence of caries and a DFT of 3.3 ± 3.7 teeth in 5-year-old children. This is in line with what was observed in the sample of Roma children in Seville studied in the present study, which had a DFT of 5.0 ± 3.6 at 6–9 years of age [25]. This rate in the Spanish population is also markedly lower (DFT of 1.28 + −2.42) in the 2020 study [23].

We believe that the fact of non-attendance of dental care is particularly important. In a study by Zhang et al. in China, the prevalence of caries was determined to be 70% in 6-year-old children [26]. Many decayed teeth were not treated as was identified in the present study, where most caries-affected teeth were not treated by the dentist.

It is important to highlight this fact, since for the ages studied in the autonomous community of Andalusia (where the city of Seville is located), there is a specific programme of free oral health care for children aged 6 to 15 years [2]. The fact that children are excluded from the health care system despite a health centre catering to their needs being located within walking distance of their place of residence, offering them free access to basic dental care, is something that deserves a thorough analysis. It would be interesting to determine the reasons for the absence of access to children’s health centre, as the rest of the Spanish population has an RI for primary dentition of 27.1% at 5–6 years of age in the aforementioned 2020 study [23].

Regarding older children, in the Western Sahara region (southern Algeria), Almerich-Silla et al. studied children aged 6–7 and 11–13 years and recorded a prevalence of caries in 47% and 63% of cases, respectively, as a well as a DFT of 0.5 teeth and a DMFT of 1.7 teeth [27]. The prevalence of caries identified in the present study is higher than that of the Oral Health Survey in Spain 2015 and 2020 [21,23].

Like the Roma children examined, other groups at risk of vulnerability in Spain have high caries rates. Nieto-García et al., in a study of the child population of Ceuta [28], reported clearly unfavorable caries rates compared to national rates in Spain. The authors reported a moderate to high prevalence of caries (DFT 3.0 ± 3.3 teeth at 7 years and DMFT 3.9 ± 3.3 teeth at 12 years), while the national results reflected a low prevalence, according to the classification proposed by the WHO [20,27] with a DFT of 1.28 + −2.42 at 5–6 years and a DMFT of 0.58 ± 1.13 at 12 years in data from the year 2020 [23].

One of the socioeconomically disadvantaged minorities classically studied are immigrants. Paredes-Gallardo et al. [29] and Almerich-Silla et al. [27] reported very high levels of dental caries at 12 years of age in the immigrant population compared to the Spanish population (71.4% vs. 40.6%; 53.2% vs. 35.3%; and 21.1% vs. 15.5% respectively). Furthermore, the study by Almerich-Silla et al. reported a DMFT at 12 years of age of 2.4 teeth in immigrants compared to 1.0 in Spaniards [30]. As we can see, there are similar differences to those reported in our study. The same occurs when we compare the results with data from refugee children, which is similar to those obtained in our study and significantly worse than that of the Spanish population in which children have been taken in for residence [31].

The restoration rate (defined as the percentage of fillings performed as a percentage of the total DMFT or DFT) is generally related to the economic power of the population, as well as the degree of information and access to health services. It is considered an indicator of the level of dental care of the population. 

Llodra-Calvo et al., in a survey of children from immigrant and Spanish families, identified that RI was higher in Spaniards (57.7%) than in foreigners (45.8%). On the other hand, Nieto-García et al., with children from Ceuta, recorded RI from 0.0 to 2.7% of teeth at 7 years and from 0.0 to 9.9% of teeth at 12 years for school children of non-active and active parents, respectively; these data suggest a relationship of RI according to parental occupation [21,28,31]. As in our study, RI is significantly higher in the national population than in the minority studied.

If we examine similar studies conducted in other countries, the high restorative needs of Roma children compared to the rest of the population follow a similar comparison when we analyse the differences between ethnic minority populations and the rest of the population. 

In the United States, in the study by Dye et al. [32], it was reported that in children aged 6 to 11 years, one in five children had experienced dental caries in their permanent teeth. The prevalence of dental caries was higher in Hispanic children compared to non-Hispanic white children or non-Hispanic Asian children.

Other studies in Spain show lower treatment needs than those detected in the studied Roma group [33].

The 2010 oral health survey shows that extraction and endodontic needs at 12 years of age were relatively small (3.5% and 1.9%, respectively) [34] compared to those identified in Roma children who participated in the present study. In the children of Navarra, an RI comparison was conducted and an increase in the index was observed with increasing age. Restoration rates (RI) were 16.7%, 48.1%, 77.3% and 78.7% [35].

In the children and youth cohorts, the exodontic needs are practically non-existent at 0.7 and 1.0% for the 12- and 15-year-old cohorts, respectively [36]. In the results of the present study, it was observed that children aged 6–9 years had on average more teeth in need of extraction and endodontics (7.0 ± 2.8) compared to the older group (3.3 ± 2.6). In children aged 10–13 years, the most prevalent treatment need was fillings (6.8 ± 2.3); this average was higher than that observed in the 6–9-year-old group (3.3 ± 2.5).

The examined school children showed high levels of gingivitis and bleeding on probing, so it is important to improve the oral hygiene conditions of the gypsy child population.

In this study, when calculating the needs of all the sextants, it was observed that all the examined children needed periodontal intervention. Most (70.4%) would require prophylaxis and training in brushing techniques. In the study group, dental visits and brushing frequency are low, which explains the high levels of periodontal tissue involvement identified in the examined Roma school children.

Other populations in developing countries also have poor oral hygiene conditions. African populations generally have poor oral hygiene and the methods used for oral cleaning may be unconventional (such as the use of chewing sticks or sponges; banana stems with charcoal powder; ashy plant leaves used with cotton wool, cloth, or fingers), leading to poor plaque removal and consequently calculus formation, even at a young age [37].

Nieto-García et al. studied a population in Ceuta, where there is a mix of populations and cultures. A total of 54% of 12-year-olds had healthy periodontium, and 18% of them presented with calculus, while 7-years-olds had healthy periodontium in 72% of cases and presented with calculus in 8% of cases [28].

In the 2015 oral health survey conducted among the 12- and 15-year-old age groups, the percentage of individuals with dental calculus was 21.7% and 28.6%, respectively, while a healthy periodontium was detected in 48.2% and 46.0%, respectively [21].

More than 70% of Roma school children require immediate dental care. There were no gender differences; this percentage is higher than that identified in Syria, where 29% of the study group experienced pain due to the presence of untreated caries. The same study also reported that 5-year-olds had a high level of care needs, more than 50% of which 42% required pain relief and only 33% had seen a dentist for preventive procedures [25].

It is important to analyse some aspects of access to health services in view of the lack of dental care observed in the group of Roma children who participated in the present study. From the point of view of health-related behaviour, health-seeking behaviour (HSB) has been defined according to Olenja as “any action or inaction taken by people who perceive that they have a health problem or are ill in order to find an appropriate remedy” [38,39].

A study in Nigeria where socio-cultural level was identified as influencing health-seeking behaviour showed that less educated workers had less appropriate health-seeking behaviour [40].

A study in African-American minorities showed that fear and lack of information can limit minority participation in clinical trials. This model has also been used in Asian minorities to understand the limitations of this group in their access to medical services [41].

Brach and Fraser developed a conceptual model of the potential of cultural competence to reduce racial and ethnic health disparities using the literature on cultural competence and disparities to lay the foundation for the model and conduct validity assessments.

The authors conclude that, while there is substantial research evidence suggesting that cultural competence should indeed work, health systems have little evidence of which cultural competence techniques are effective and less evidence on the time and ways to implement them appropriately [42].

In the case of the Roma ethnic group in Seville, it could be relevant to improve the competencies of health workers in aspects related to this ethnic group in such a way that the type of behaviour of this community is understood and communication channels are identified that allow the health worker to provide a clear message and consider the cultural diversity that allows the Roma community to make a more appropriate use of health services that result in better quality of life and respect for their traditions in harmony with the society that houses them.

There is information on the application of this model in the search for strategies to improve oral hygiene in children [43]. The understanding of children's behavioural decisions to practice good oral hygiene is considered to be limited.

A study conducted in the United States interviewed eight focus groups of 42 children (second to fifth grade) about their caries history, perceived confidence in brushing, perceived susceptibility and vulnerability to caries and poor oral health, and perceived benefits and barriers to oral hygiene practice.

The results of the focus group led to the conclusion that emphasis should be placed on the positive aspects of maintaining good oral hygiene due to its contribution to appearance and its implications for a general healthy body and self-image [43].

An important element of the model is self-efficacy. This construct has been used in studies on oral hygiene and the ability of parents to maintain good hygiene habits in their children [44].

This study showed that the number of children brushing twice a day also increased significantly with increasing level of guardian self-efficacy (OR: 2.14, 95% CI: 1.91 to 2.41).

Brushing behaviour was associated with the self-efficacy of parents or guardians in convincing their children to brush twice a day. The importance of behavioural interventions to improve oral health has been identified in Greek children [45].

## 5. Conclusions

The prevalence and rates of caries in the two age cohorts of Roma children studied are particularly high and much higher than in the rest of the Spanish population. This demonstrates the need to design specific health promotion and oral disease prevention campaigns for this disadvantaged population.

## Figures and Tables

**Table 1 healthcare-11-01016-t001:** Sociodemographic description of the study sample (*n* = 108) according to age groups.

	6–9 Years Old (*n* = 52; 48.1%)		10–13 Years Old (*n* = 56; 51.9%)	
**Area of** **residence**	** *N* **	**%**	** *n* **	**%**
Sevilla Sur	45	86.5	43	76.8
Sevilla Norte	7	13.5	13	23.2
Gender				
Boy	26	50	28	50
Girl	26	50	28	50
Age	Mean	SD	Mean	SD
	7.8	1.1	11.2	1.0
**Occupation of the head of the household**	** *N* **	**%**	** *N* **	**%**
Peddler	46	88.5	41	73.2
Laborer	2	3.8	0	0
Construction	1	1.9	2	3.6
Housekeeping	1	1.9	4	7.1
Unemployment	1	1.9	9	13.1
Painter	1	1.9	0	0

**Table 2 healthcare-11-01016-t002:** Clinical description of caries status in the sample according to age groups.

	6–9 Years		10–13 Years		Total	
Temporal teeth	Mean	SD	Mean	SD	Mean	SD
Decayed	5.0	3.1	1.3	2.0	3.1	3.2
Filled	0.0	0.0	0.0	0.0	0.0	0.0
Missing	0.0	0.0	0.0	0.0	0.0	0.0
DFT	5.0	3.1	1.3	2.0	3.1	3.2
RI (Restauration index) (temporal %)	0.0	0.0	0.0	0.0	0.0	0.0
Permanent teeth						
Decayed	1.2	1.4	4.5	3.3	2.9	3.1
Filled	0.0	0.0	0.0	0.3	0.1	0.3
Missing	0.0	0.0	0.1	0.5	0.0	0.2
DMFT	1.2	1.4	4.6	3.3	3.0	3.1
RI (permanents %)	0.0	0.0	1.0%	7.0	0.6%	5.6
All teeth *						
RI total	0.0	0.0	1.0	6.9	0.5	5.0
SiC **	10.1	1.5	10.4	2.0	10.2	1.7

* We applied the percentage of fillings to the total number of primary and permanent teeth. ** Caries significance index.

**Table 3 healthcare-11-01016-t003:** Description of dental treatment needs according to age groups.

	6–9 Years		10–13 Years		Total	
Specific requirements	Mean	SD	Mean	SD	Mean	SD
Fillings	3.3	2.5	6.8	2.3	4.9	2.0
Invasive treatment (extraction or endodontics)	7.0	2.8	3.3	2.6	5.1	1.1
Distribution of filling requirements						
	*n*	%	*N*	%	*n*	%
None	12	23.1	7	12.5	19	17.6
1–2 fillings	21	40.4	9	16.1	30	27.8
3 or more	19	36.5	40	71.4	59	54.6
Distribution of needs for endodontics or extractions (invasive treatment)						
None	11	21.2	14	25.0	25	23.1
1–2 treatments	7	13.5	29	51.8	36	33.3
3 or more	34	65.4	13	23.2	47	43.5

**Table 4 healthcare-11-01016-t004:** Description of clinical and subjective variables for the 12- and 13-year-old age group.

	Group of 12- and 13-year-olds	
	*N*	%
Malocclusion		
No	6	22.2
Mild	4	14.8
Moderate/Severe	17	63.0
Degree of MIH		
No	16	59.3
Mild	6	22.2
Moderate	4	14.8
Severe	1	3.7
Fluorosis		
No	27	100.0
Pain or discomfort in the last 12 months		
Never	4	14.8
Hardly ever	4	14.8
Sometimes	6	22.2
Quite often	12	44.4
Very often	1	3.7
Eating problems in the last 12 months		
Never	11	40.7
Hardly ever	4	14.8
Sometimes	9	33.3
Quite often	3	11.1
Very often	0	0.0

**Table 5 healthcare-11-01016-t005:** Frequency of brushing by sex in the age group of 10–13 years.

Frequency of Toothbrushing	Boys		Girls		*p* Value
	*n*	%	*n*	%	
Never	22	40.7	4	7.4	0.031 *
<Once per week	26	48.1	18	33.3	
Sometimes a week	3	5.6	18	33.3	
Once per day	3	5.6	4	7.4	
2–3 times per day	0	0	10	18.6	

* *p* < 0.05.

**Table 6 healthcare-11-01016-t006:** Comparison of dental treatment needs by sex in different age groups.

	6–9 Years (*n* = 52; 48.1%)					10–13 Years (*n* = 56; 51.9%)				
Need of:	Boy (*n* = 26)		Girl (*n* = 26)		*p* value	Boy (*n* = 28)		Girl (*n* = 28)		*p* value
	M	SD	M	SD		M	SD	M	SD	
Fillings	1.5	1.4	2.9	2.4	0.034 *	4.2	2.8	3.8	2.5	0.532
Tooth extractions	3.0	3.3	3.2	3.0	0.654	1.3	1.3	1.8	2.1	0.628
Invasive treatments (endo + exo)	4.0	3.4	4.3	3.1	0.692	1.6	1.4	2.1	2.3	0.788

* *p* < 0.05.

**Table 7 healthcare-11-01016-t007:** Comparison of dental visits by sex in the sample.

Visits in the Last Year	Boys		Girls		*p* Value
	*n*	%	*n*	%	
No	7	13.0	9	16.7	0.681
Once	11	20.7	18	33.3	
Twice	7	13.0	9	16.7	
3 times	0	0.0	4	7.4	
Has never gone	29	53.3	14	25.9	

## Data Availability

Data are available by corresponding authors.

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
