# Peer review of "Oral Health Status in a Group of Roma Children in Seville, Spain"

_healthcare, 2023, doi:10.3390/healthcare11071016_

Round 1

Reviewer 1 Report

Thank you for the paper.  

1. Overall, the paper is relevant. 

2. Consider including definition of Social Determinants of Health 

3. Define health equity and why is the importnat 

4. Sample size was caluclated based on permanent caries rate but the children assesses were in mixed dentition stage, explain why was this method chosen 

5. Explain- what teeth were determined carious, non carious

6. How was conclusion reached without radiographs what teeth needed RCT and what teeth needed extractions 

7. Was the bleeding of gums self reported or elicited or spontaenous?

8. How was peridontal need defined 

Thank you

Author Response

ANSWER TO REVIEWER 1.

Thank your for your comments. We will correct everything you´ve suggested. We will write them in red colour added in the main text. 

  1. Consider including definition of Social Determinants of Health

The social determinants of health (SDH), in accordance with WHO, are the non-medical elements that affect health outcomes. In addition to the larger group of factors and systems influencing the conditions of daily life, these are the circumstances in which people are born, develop, work, live, and age. These factors and systems include political systems, societal norms, social policies, economic policies and systems, and development objectives. 

Health inequalities, or the unfair and preventable variations in health status seen within and between regions, groups of population or even countries, are significantly influenced by SDH. Health and sickness follow a social gradient across these regions of all income levels: the worse one's socioeconomic standing, the worse one's health. 

  1. Define health equity and why is the importnat

Health equity, as defined by the WHO, is the absence of unjust and preventable or redressable disparities in health among population groups classified according to social, economic, demographic, or geographic factors. 

  1. Sample size was caluclated based on permanent caries rate but the children assesses were in mixed dentition stage, explain why was this method chosen

We have rewritten the part corresponding to the sample size calculation. 

  1. Explain- what teeth were determined carious, non carious

Thank you for your suggestion, we have added a paragraph explaining that we use the WHO recommendations in their book Oral Health Surveys, Basic Methods 5th edition. 

  1. How was conclusion reached without radiographs what teeth needed RCT and what teeth needed extractions

Thank you for your suggestion. It is true that since we did not take X-rays we could not know if it was necessary to perform endodontics or extraction on the tooth because this field work was not performed under the same conditions as in an ordinary dental office. For this reason we have included that the teeth with caries with evident pulp involvement (we have reflected it as invasive treatment) have been unified in a single item. 

  1. Was the bleeding of gums self reported or elicited or spontaenous?

Following the guidelines advised by the WHO for oral health surveys in the aforementioned book and taking into account the criteria of the Community Periodontal Index (CPI), bleeding (when it is included as such in the index) is bleeding on probing in the periodontal examination. 

Reviewer 2 Report

Thank you for allowing me to review this manuscript, whose purpose was to identify the oral health status of the gipsy 6 to 13 years old children from Seville.

The manuscript's topic is interesting, but many things could be improved.

1. Abstract - the introduction should be shorter and more appropriate. Immediately move on to the objectives of the research.

2. The article has Spanish words (e.g. in the introduction raison d'être).

3. The introduction is dull and uninteresting. It does not introduce the problems of oral health. Lines 33 to 55 are unnecessary; the rest should be formatted differently.

4. The last sentence of the introduction needs to be shorter and more precise, so you should ask yourself what the purpose of the work is.

5. Who approved the study? At what time was it carried out?

6. How was the sample size calculated? Based on which study, where is the reference?

7. State whose statistics package it is. Please provide the correct test names.

8. What does media mean in table one? Remove the percent sign next to the number.

9. What does this sentence mean at the beginning of the results (Socio-demographic data and oral health-related habits (frequency of toothbrushing and dental visits during the most recent year) were collected.)

10. Where the results are the student t-test and chi-square test, I don't see any p-value in the results.

11. What is the study's strength, and what are the limitations?

12. The conclusions are too general and do not fully follow the research topic.

13. The literature is not written following the magazine's instructions.

Author Response

REVIEWER 2. 

Thank you for your commentos in order to improve the paper. We have corrected all your suggestions and put in another colour in the main text. 

  1. Abstract - the introduction should be shorter and more appropriate. Immediately move on to the objectives of the research.

We have rewriten the introduction 

  1. The article has Spanish words (e.g. in the introduction raison d'être).

That was an error. We have eliminated it. 

  1. The introduction is dull and uninteresting. It does not introduce the problems of oral health. Lines 33 to 55 are unnecessary; the rest should be formatted differently.

Ok, we have re-writen it. 

  1. The last sentence of the introduction needs to be shorter and more precise, so you should ask yourself what the purpose of the work is.

Ok, we have re-writen it. 

  1. Who approved the study? At what time was it carried out?

The study is supported by and is part of the actions of the social dentistry foundation (FOS) with links to the University of Seville, as well as the rest of the public administrations of Andalusia. 

It was carried out in the last week in September 2019.  

  1. How was the sample size calculated? Based on which study, where is the reference?

The study by Bravo et al.2015 has been included as a reference study when considering the sample size.  

  1. State whose statistics package it is. Please provide the correct test names.

SPSS 21.0 version was used. 

  1. What does media mean in table one? Remove the percent sign next to the number.

It was an error, it´s been corrected. 

  1. What does this sentence mean at the beginning of the results (Socio-demographic data and oral health-related habits (frequency of toothbrushing and dental visits during the most recent year) were collected.)

We have eliminated the sentence. 

  1. Where the results are the student t-test and chi-square test, I don't see any p-value in the results.

We have added the p value in the results where it was precised. 

  1. What is the study's strength, and what are the limitations?

We have included the limitations of this cross-sectional study. 

  1. The conclusions are too general and do not fully follow the research topic.

We have re writen the conclusions. 

Reviewer 3 Report

The manuscript investigates an interesting topic. The manuscript is fulfilling the requirements of the journal. However, the study population is quite small. 
This must be stated in the manuscript that the limitation of the study was the low number of participants and further studies needed to follow up.

I would also include that social dentistry is efficient if the prevention and education campaigns are existing and are provided for the people who are in need.

There is barely any explanation about what kind of validation was with the study questionnaire. Please include all the questions in supplementum or in a table. The reason for it, is that it must be validated, or extracted from an existing pool (WHO Questionnaire on oral health ANNEX7...)

Please also refer to the following publications:

1. https://pubmed.ncbi.nlm.nih.gov/34509286/

2. https://pubmed.ncbi.nlm.nih.gov/35010805/

Author Response

REVIEWER 3. 

The manuscript investigates an interesting topic. The manuscript is fulfilling the requirements of the journal. However, the study population is quite small.  
This must be stated in the manuscript that the limitation of the study was the low number of participants and further studies needed to follow up. 

Thank you for your words and suggestions for improving the article.  

The sample size was calculated on the basis of the prevalence of caries studied in the Spanish population, which is why this sample size was arrived at. It is true that for higher statistical power, the number of patients seen could have been increased. 

I would also include that social dentistry is efficient if the prevention and education campaigns are existing and are provided for the people who are in need. 

Thank you, we have included this concept in the text at the Introduction and in the conclusions. 

There is barely any explanation about what kind of validation was with the study questionnaire. Please include all the questions in supplementum or in a table. The reason for it, is that it must be validated, or extracted from an existing pool (WHO Questionnaire on oral health ANNEX7...) 

The questionnaire is validated by WHO. We have used the questionnaire of the book Oral Health Surveys. Basic Methods 5th edition, and it´s Annex 8 the validated questionnaire used. We´ve inserted this in tha m&M part. 

Round 2

Reviewer 1 Report

Thank you for the revisions. 

One minor revision suggestion- Please revise line 57. 

The so-called first world also presents situations and groups of clear social exclusion. 57 In Massachusetts, through Harvard University, actions are carried out with families with 58 low-income power, homeless people

Reviewer 2 Report

I still do not see strengths of the study and literature written according Journal instructions. 
